# Hormone Receptor Expression and Activity for Different Tumour Locations in Patients with Advanced and Recurrent Endometrial Carcinoma

**DOI:** 10.3390/cancers16112084

**Published:** 2024-05-30

**Authors:** Maartje M. W. Luijten, Willem Jan van Weelden, Roy I. Lalisang, Johan Bulten, Kristina Lindemann, Heleen J. van Beekhuizen, Hans Trum, Dorry Boll, Henrica M. J. Werner, Luc R. C. W. van Lonkhuijzen, Refika Yigit, Camilla Krakstad, Petronella O. Witteveen, Khadra Galaal, Alexandra A. van Ginkel, Eliana Bignotti, Vit Weinberger, Sanne Sweegers, Ane Gerda Z. Eriksson, Diederick M. Keizer, Anja van de Stolpe, Andrea Romano, Johanna M. A. Pijnenborg

**Affiliations:** 1Department of Obstetrics and Gynaecology, Radboud University Medical Centre, 6525 GA Nijmegen, The Netherlandshanny.ma.pijnenborg@radboudumc.nl (J.M.A.P.); 2Department of Gynaecology, Rijnstate Hospital, 6815 AD Arnhem, The Netherlands; 3Department of Obstetrics and Gynaecology, Canisius Wilhelmina Hospital, 6532 SZ Nijmegen, The Netherlands; 4GROW-School of Oncology and Developmental Biology, Maastricht University Medical Center+, 6229 ER Maastricht, The Netherlands; 5Department of Pathology, Radboud University Medical Center, 6525 GA Nijmegen, The Netherlands; 6Division of Medicine, Department of Gynecological Oncology, Oslo University Hospital, 0424 Oslo, Norway; 7Faculty of Medicine, Institute of Clinical Medicine, University of Oslo, 0372 Oslo, Norway; 8Department of Gynecologic Oncology, Erasmus MC Cancer Institute, Erasmus Medical Center Rotterdam, 3015 GD Rotterdam, The Netherlands; h.vanbeekhuizen@erasmusmc.nl; 9Center for Gynecologic Oncology Amsterdam, Netherlands Cancer Institute, 1066 CX Amsterdam, The Netherlands; 10Department of Gynaecology, Catharina Hospital, 5623 EJ Eindhoven, The Netherlands; dorry.boll@catharinaziekenhuis.nl; 11Department of Obstetrics and Gynecology, Maastricht University Medical Center+, 6229 HX Maastricht, The Netherlands; erica.werner@mumc.nl; 12Department of Gynaecology and Obstetrics, Amsterdam University Medical Centers, University of Amsterdam, 1105 AZ Amsterdam, The Netherlands; l.r.vanlonkhuijzen@amsterdamumc.nl; 13Department of Obstetrics and Gynecology, University Medical Center Groningen, 9713 GZ Groningen, The Netherlands; r.yigit@umcg.nl; 14Department of Gynecology and Obstetrics, Haukeland University Hospital, 5009 Bergen, Norway; camilla.krakstad@uib.no; 15Department of Medical Oncology, University Medical Center Utrecht, 3584 CX Utrecht, The Netherlands; p.o.witteveen@umcutrecht.nl; 16Sultan Qaboos Comprehensive Cancer Center, Muscat P.O. Box 566 PC 123, Oman; 17Division of Obstetrics and Gynecology, A. Nocivelli Institute for Molecular Medicine, ASST Spedali Civili di Brescia, 25123 Brescia, Italy; 18Department of Obstetrics and Gynecology, Faculty of Medicine, Masaryk University, 625 00 Brno, Czech Republic; 19Department of Obstetrics and Gynecology, University Hospital Brno, 625 00 Brno, Czech Republic; 20InnoSIGN, 5656 AE Eindhoven, The Netherlands; 21DCDC-Tx B.V., 5263 EM Vught, The Netherlands; anja.van.de.stolpe@dcdc-tx.com

**Keywords:** endometrial cancer, hormone receptor, tumour location

## Abstract

**Simple Summary:**

Oestrogen and progesterone are two sex hormones that are important in the development of cancer of the inner lining of the uterus and endometrial cancer (EC). Oestrogen binds to the oestrogen receptor (ER) and progesterone to the progesterone receptor (PR). The presence of these receptors is important because the response to hormonal therapy is higher when the receptors are present. However, as EC grows and spreads throughout the body, ER and PR may be lost. In this study, we found that tumours that have spread to other organs throughout the blood and tumours that have spread in the abdomen have a relatively high presence of ER/PR, while tumours located in the lymph nodes have a lower presence of PR. ER/PR-IHC were not lower in tumours that had previously been treated with radiotherapy. This might influence the application of hormonal therapy in the future.

**Abstract:**

Background: Response to hormonal therapy in advanced and recurrent endometrial cancer (EC) can be predicted by oestrogen and progesterone receptor immunohistochemical (ER/PR-IHC) expression, with response rates of 60% in PR-IHC > 50% cases. ER/PR-IHC can vary by tumour location and is frequently lost with tumour progression. Therefore, we explored the relationship between ER/PR-IHC expression and tumour location in EC. Methods: Pre-treatment tumour biopsies from 6 different sites of 80 cases treated with hormonal therapy were analysed for ER/PR-IHC expression and classified into categories 0–10%, 10–50%, and >50%. The ER pathway activity score (ERPAS) was determined based on mRNA levels of ER-related target genes, reflecting the actual activity of the ER receptor. Results: There was a trend towards lower PR-IHC (33% had PR > 50%) and ERPAS (27% had ERPAS > 15) in lymphogenic metastases compared to other locations (*p* = 0.074). Hematogenous and intra-abdominal metastases appeared to have high ER/PR-IHC and ERPAS (85% and 89% ER-IHC > 50%; 64% and 78% PR-IHC > 50%; 60% and 71% ERPAS > 15, not significant). Tumour grade and previous radiotherapy did not affect ER/PR-IHC or ERPAS. Conclusions: A trend towards lower PR-IHC and ERPAS was observed in lymphogenic sites. Verification in larger cohorts is needed to confirm these findings, which may have implications for the use of hormonal therapy in the future.

## 1. Introduction

Endometrial cancer (EC) is the most frequent malignancy of the female genital tract in industrialised countries [1]. The incidence of EC is increasing due to risk factors such as prolonged life expectancy and obesity [2]. Most women are diagnosed with EC at an early stage, resulting in a favourable prognosis with a 5-year survival rate of 80–90% [3,4]. However, a minority of patients diagnosed with advanced or metastatic EC have a significantly worse outcome, with a 5-year survival rate of only 19% [4]. Up to 20% of patients will develop local recurrence or metastatic disease [3,4,5,6,7]. Curative treatment options for metastatic EC are limited to women who only have locoregional metastases or isolated metastases [1,8,9]. For women with more advanced disease, only palliative options remain and are limited to the following: chemotherapy and hormonal therapy (HT) or novel therapies, including immunotherapy and other targeted treatment options [9,10,11].

Both response rate (RR) and toxicity are important in determining the most suitable systemic therapy for a patient. Chemotherapy with carboplatin/paclitaxel yields an RR of 60% and a progression-free survival of about 13 months. However, it is associated with serious toxicity in at least 35% of patients [1,12,13,14,15,16]. The addition of immunotherapy to chemotherapy in women with mismatch repair-deficient tumours increased for the 12-month progression-free survival from 38% to 74% [17]. However, toxicity was also increased compared with the chemotherapy alone group [17]. Hormonal therapy shows an RR of 20–40% in unselected patients, but in responders, it can provide a durable effect lasting several years with minimal side effects [18,19,20,21,22,23].

Unopposed oestrogen exposure plays a pivotal role in EC development. In the normal endometrium, as well as in endometrial cancer, progesterone counteracts oestrogen-induced proliferation and tumour growth. However, it is currently not completely clear if oestrogen is responsible for tumour growth in all ECs, making the application of progesterone potentially ineffective. Immunohistochemical expression of the oestrogen receptor and progesterone receptor (ER/PR-IHC) is a well-established predictive biomarker for response to hormonal therapy, with the highest RR in patients expressing both ER and PR [19,23,24,25,26]. However, ER-IHC does not necessarily reflect ER activity [27,28]. Besides this, receptor status is still not routinely assessed prior to treatment, and there is no consensus on the optimal cut-off value for hormone receptor positivity (ER/PR) in EC [20,21,29]. In addition, the molecular classification has further challenged the position of hormonal therapy [30,31,32,33].

The Prediction of Response to Hormonal therapy in advanced and recurrent Endometrial Cancer (PROMOTE) study showed an association between the ER, PR and ER pathway activity (ERPAS) and the outcome [24,28]. The response was only observed in patients with a pre-treatment tumour expression of ER/PR > 50%. In the group with a tumour expression of PR > 50% and ERPAS > 15 an RR of 57.6% was achieved. However, there was considerable variability among the samples in terms of tumour location and expression profile.

Primary well-differentiated tumours are typically sensitive to hormonal therapy; however, they may lose their hormone sensitivity over time andin response to applied adjuvant treatment such as radiotherapy and/or chemotherapy [34,35,36,37,38,39]. Changes in hormone receptor expression from primary tumours to metastases have been reported in up to 50% of cases [35,36,37]. As metastases often exhibit heterogeneity compared to the primary tumour, the reassessment of hormone receptor status within metastases is recommended [23,34,35,36]. The analysis of ER/PR-IHC expression and ERPAS at different tumour locations can improve the understanding of cancer spread and help refine patient selection for hormonal therapy [33]. Therefore, we evaluated ER/PR-IHC expression and ERPAS between different tumour locations in advanced and recurrent EC collected in the PROMOTE-study [24].

## 2. Materials and Methods

### 2.1. Design and Setting

The current study used samples collected from patients in the PROMOTE-study [24]. Patients were included if they were treated with any type of hormonal therapy for advanced or recurrent EC and an available pretreatment tumour biopsy was required. Concurrent therapy was not permitted, and a minimum of 6 months of documented follow-up after the initiation of hormonal treatment was required. Patients receiving hormonal therapy for other indications, patients receiving adjuvant hormonal therapy, and patients with endometrial (stromal) sarcoma were excluded. All participating centres obtained approval from the institutional review board or national ethics committee approval and obtained patient consent according to local regulations. The Radboudumc Institutional Review Board approval number was 2017-3803.

### 2.2. Patients

Baseline characteristics, including the patient characteristics age, BMI, and the tumour characteristics histology and grade (grade 1–2: low grade; grade 3: high grade), FIGO stage, and biopsied tumour location, were retrieved from patient records. Primary surgical treatment was classified as hysterectomy with bilateral salpingo-oophorectomy, with or without lymph node dissection. Some patients received (adjuvant) radiotherapy or chemotherapy. Radiotherapy could include either vaginal brachytherapy (VBT) and/or external beam radiotherapy (EBRT). All types of hormonal therapy were included (progestin, tamoxifen, aromatase inhibitor). Details of the hormonal therapy used, including type and dosage, were documented.

### 2.3. Tumour Locations

All biopsied tumour sites were classified according to the hypothesised route of cancer spread based on the study by Kurra et al. [40] as follows:Uterine biopsies from patients with advanced EC (cervix, uterus);Lymphogenic tumour location (inguinal lymph node, para-aortic lymph node, pelvic lymph node, supraclavicular lymph node);Hematogenous tumour location (liver, lung);Intra-abdominal tumour location (colon, peritoneum, omentum);Port-site tumour location (surgical scar, umbilical cord);Vaginal vault tumour location from recurrent EC patients.

### 2.4. Immunohistochemical Staining and Scoring

The immunohistochemical (IHC) analysis of ER and PR expression was performed on 4 µm tumour-containing sections of formalin-fixed paraffin-embedded (FFPE) tumour blocks [27]. Additional details, including the antibodies used, are provided in Appendix A [24]. Two investigators (WvW and JB) independently assessed the percentage of tumour cells expressing nuclear ERα and PR (Figure 1). In the case of a disagreement, the final score was determined in a consensus meeting. Both reviewers were blinded to the clinical and pathological data.

#### 2.4.1. RNA Isolation

The tissue of interest was micro- or macrodissected from two consecutive 10 µm FFPE sections. Subsequently, RNA was then extracted using the miRNeasy FFPE kit (Qiagen, Hilden, Germany) according to the manufacturer’s instructions and as fully described in Appendix A.

#### 2.4.2. ER Pathway Activity Test

A knowledge-based Bayesian computational model, previously shown to have predictive value in breast cancer, was used to assess ER pathway activity, as outlined in Appendix A. [24]. Briefly, the model utilises the mRNA expression levels of ER target genes measured in tissue samples, estimated by InnoSIGN using OncoSIGNal^TM^ (Philips Molecular Pathway Diagnostics, Eindhoven, The Netherlands) to infer the probability of a transcriptionally active ER transcription. A score of 0 indicates the lowest odds of pathway activity, while 100 represents the maximum odds inferred by the model.

Institutional review boards at all participating centres approved the study prior to initiation. In accordance with the code of conduct for the responsible use of human tissue in medical research, patient consent was not required for this study [41].

### 2.5. Outcome Measures

The primary outcome was defined as ER-IHC and PR-IHC expression and ERPAS according to the route of cancer spread [40]. In the PROMOTE-study, cut-offs for ER-IHC, PR-IHC and ERPAS were defined based on response. This resulted in three different categories for ER-IHC and PR-IHC: ≤10%, >10–50% and >50%. For ERPAS, this resulted in two different categories: ≤15 and >15. [24]

### 2.6. Statistical Analysis

Clinical and pathological parameters of the different tumour locations were compared using Pearson χ^2^ for categorical variables, and the Kruskal–Wallis test was used for continuous variables due to non-normal distribution of age, BMI, ER-IHC, PR-IHC and ERPAS in all subgroups, assessed by the Shapiro–Wilk test. Post hoc analysis with Bonferroni correction was performed for significant differences identified by the Chi-square test.

Differences in receptor status and pathway activity between uterine biopsies and other locations, as well as between lymphogenic and other locations, were assessed using Fisher’s exact test with cut-offs of 10% and 50% for ER-IHC and PR-IHC and a cut-off of 15 for ERPAS due to low expected cell counts.

Fisher’s exact test was used to compare immunohistochemistry and ERPAS between the uterine biopsies of advanced EC and metastasis/recurrence at different locations.

Differences in receptor status and pathway activity between the samples with and without prior applied radiotherapy in the vaginal vault samples were analysed using the independent samples t-test or the Mann–Whitney U test, depending on subgroup normality, assessed by the Shapiro–Wilk test.

All tests were two-tailed, and *p*-values of <0.05 were considered significant. SPSS (version 25.0 for Microsoft, SPSS Inc., Chicago, IL, USA) was used for statistical analyses.

## 3. Results

### 3.1. Patients

A total of 105 eligible patients were included in the PROMOTE-study (Figure 2). Due to the exclusion of cases with insufficient tumour tissue (*n* = 15), biopsies during HT (*n* = 1), stage I/II EC (*n* = 1), incomplete IHC analysis (*n* = 3), and HT were given to sites other than the biopsied site (*n* = 5), and a total of 80 samples were included for analysis. In two patients with recurrent EC, two different biopsies from two different sites were included.

Clinicopathological data of the included patients (*n* = 80) are shown in Table 1. In total, 54 (67.5%) patients were treated for recurrent EC and 26 (32.5%) for advanced EC. The mean age of all patients was 71.9 years (SD = 9.6), and the mean body mass index (BMI) was 30.4 (SD = 7.8). Most patients (*n* = 61, 82.4%) were diagnosed with grade 1–2 tumours, and 13 were diagnosed (16.7%) with grade 3 tumours.

### 3.2. Samples

The tumour biopsies are classified according to the route of cancer spread, illustrated in Figure 3. Within all tumour locations, 12 biopsies (15%) were uterine biopsies from patients with advanced EC. The remaining 78 biopsies (85%) were derived from tumour locations with advanced stage (*n* = 14) or recurrent EC (*n* = 54) with 14 hematogenous (17.5%), 12 lymphogenic (15.0%), and 9 intra-abdominal (11.3%). In addition, 5 biopsies were port-site (6.3%), and 28 were vaginal vault biopsies (35.0%) (*exclusively recurrent EC*). A total of 33 biopsies were taken from patients who had previously received radiotherapy, either vaginal brachytherapy (VBT) and/or external beam radiation therapy (EBRT). Sixteen biopsies were taken outside of the radiation fields. As a result, 17 biopsies taken from locations within the previously irradiated field (vaginal vault biopsies: *n* = 16, lymphogenic location: *n* = 1) were included in this analysis. A total of 63 biopsies from non-irradiated sites were included.

### 3.3. Expression of ER/PR-IHC and ER Pathway in Relation to Pattern of Spread

Within the cohort, ER-IHC expression was assessed in 77 biopsies, with PR-IHC in 78 biopsies and ERPAS analysis in 72 biopsies. Within all biopsies, the mean ER-IHC expression was 77.3% (*n* = 77), PR-IHC 49.5% (*n* = 78) and ERPAS 16.2% (*n* = 72) (Table 1). The results of the ER/PR-IHC and ERPAS analysis are illustrated in Figure 4.

The ER-IHC expression was generally high in both uterine and metastatic locations (Figure 4A). An absence of ER-IHC expression was found in three cases of hematogenous, lymphogenic and intra-abdominal biopsies, respectively. Lymphogenic locations showed the highest proportion of ER-IHC expression <50%, yet differences were not significant.

PR-IHC expression was high in uterine biopsies, with 75.0% (*n* = 9/12) of cases showing >50% expression (Figure 4B). Among the other locations, lymphogenic locations had the highest proportion of <50% PR-IHC expression, but this was not significantly different from all other biopsies combined (*p* = 0.074). Intra-abdominal and hematogenous locations had the highest percentage of PR-IHC > 50% (71.4%, *n* = 5/7 and 69.2%, *n* = 9/13, respectively).

A total of 58.3% of uterine biopsies were ERPAS > 15 (Figure 4C). Lymphogenic sites were ERPAS-positive in 27.3% (*n* = 3/11) with port-site locations in 20.0% (*n* = 1/5). This was lower compared to other locations, although not significant (*p* = 0.294). ERPAS > 15 was the highest in intra-abdominal (71.4%, *n* = 5/7) and hematogenous locations (60%, *n* = 6/10).

### 3.4. Expression of ER/PR-IHC and ER Pathway Activity in Relation to Tumour Grade

ER-IHC and PR-IHC expression was stratified by grade in all non-uterine locations (*n* = 63/74, 85.1%), as shown in Figure 5A,B. There were no differences in ER/PR-IHC expression between low- and high-grade EC. However, the numbers for high grades were relatively low (ER-IHC *n* = 7/60, PR-IHC *n* = 7/61). The ERPAS was comparable for low- and high-grade tumours, ERPAS > 15 in 50.0% (*n* = 24/48) and 28.6% (*n* = 2/7), respectively (X^2^
*p* = 0.289) (Figure 5C).

Although not significant, tumour biopsies from the lymphogenic locations tended to have higher-grade tumours, with 36.4% (*n* = 4/11) having grade 3 tumours, of which only 50% (*n* = 2/4) had PR-IHC > 50% and 33.3% (*n* = 1/3) had ERPAS > 15 (Appendix A).

Analysis of endometrioid endometrial cancer (EEC) versus non-endometrioid endometrial cancer (NEEC) was not possible due to the small number of NEEC biopsies (*n* = 4/80, 5.0%).

### 3.5. Previous Radiotherapy and IHC

As shown in Figure 5, there was no difference in ER-IHC between samples with and without prior RT (Figure 6A). PR-IHC > 50% occurred in 43.8% of samples from previously irradiated samples (*n* = 7/16) compared to 61.3% of non-irradiated samples (*n* = 38/62) (X^2^
*p* = 0.329) (Figure 6B). Similarly, ERPAS > 15 was present in 43.8% of previously irradiated samples (*n* = 7/16) compared to 51.8% of non-irradiated locations (*n* = 29/56) (X^2^
*p* = 0.571) (Figure 6C). These differences were not significant.

Figure 6: Radiotherapy at tumour locations with and without prior RT and ER/PR-IHC and ERPAS.

## 4. Discussion

The primary objective of this study was to evaluate ER/PR-IHC expression and ER pathway activity (ERPAS) in different tumour locations in a cohort of women treated with hormonal therapy [24]. Overall, ER/PR-IHC expression was high at different tumour locations. Interestingly, lymphogenic locations showed the lowest PR-IHC expression and ERPAS activity of all locations, although these differences were not significant. There was no difference in ER/PR-IHC expression or ERPAS between low- and high-grade EC. Finally, prior radiotherapy did not affect ER/PR-IHC or ERPAS results.

Previous studies have shown that ER-IHC and PR-IHC are lower in metastases than in primary tumours [36,42]. In one study, PR-IHC < 10% occurred significantly more commonly in EEC metastases than in primary tumours, occurring in 65% and 14% of tumours, respectively [35]. In non-endometrioid endometrial cancer (NEEC), PR-IHC loss was seen in 65% of primary tumours and 94% of metastatic tumours [35]. Finally, the loss of ER/PR-IHC in pre-operative endometrial biopsies has been reported to be associated with lymphogenic tumour spread [5]. The distant metastases of EECs showed significantly lower ER-IHC expression than intra-abdominal metastases [36,42]. Although metastases may be heterogeneous compared to the primary tumour, 75% of patients with multiple metastatic lesions show homogeneous PR-IHC expression in all metastases, whereas 25% of cases show the heterogeneous loss of PR-IHC in at least one lesion [35].

### 4.1. Metastatic Locations and Immunohistochemical Expression

In our study, a relatively high overall percentage of ER/PR-IHC was present in metastatic sites. Lymphogenic locations appeared to have lower PR-IHC expression and ERPAS, whereas port-site locations showed average levels of PR-IHC > 50% and ERPAS > 15% was very low. On the other hand, hematogenous and intra-abdominal locations appeared to have higher PR-IHC and ERPAS. Since no differences in the proportions of endometrioid and non-endometrioid, or in low-grade and high-grade tumours between these groups, were found, we hypothesise that differences in PR-IHC and ERPAS may reflect different pathways of tumour progression and metastasis.

Loss of PR-IHC during cancer progression is known to correlate with markers of aggressive disease and predicts poor survival [35,43]. Progesterone counteracts oestrogen-induced effects, such as the promotion of proliferation and inhibition of apoptosis and the modulation of the tumour suppressor function. Due to its strong tumour suppressor function, the loss of PR-IHC has been shown to play a role in epithelial-to-mesenchymal-transition (EMT) [44]. EMT is an important driver of cancer progression as it increases the invasive and migratory capacity of cancer cells [43,44,45]. In non-progressive EC, intact progesterone signalling appears to be an important factor in stimulating immunosuppression and inhibiting EMT [45]. Furthermore, the lower expression of PR-IHC in metastases compared to the corresponding primary tumour suggests a role for PR-IHC loss in EMT [35,42].

The activity of the ER pathway, which has been extensively studied in breast cancer, is relevant as it is associated with ER/PR expression by analysing the downstream RNAs of ER-related target genes from which the actual oestrogen-induced tumour growth can be inferred [27,28]. However, only a subset of ER-positive tumours actually has high ER pathway activity, and levels of ER-IHC and ER pathway activity are poorly correlated [27,28]. Therefore, the ER-IHC is not a sufficiently specific biomarker of functional ER pathway activity [27,28]. By contrast, the ER pathway activity has shown a good correlation with PR-IHC, suggesting an association between PR negativity and the inactivation of oestrogen-responsive genes in tumour progression [24]. Furthermore, ERPAS was found to be lower in patients with stage 1 EEC who developed a recurrence compared to stage 1 EEC cases without a recurrence, suggesting an association between ERPAS inactivation and tumour progression [24,27,46]. Similar to breast cancer, low ERPAS is associated with adverse outcomes [32].

We observed the lowest PR-IHC and ERPAS expression within lymphogenic and port-site locations, in line with the literature, suggesting an association with EMT. Although not significant, tumour biopsies from the lymphogenic locations tended to have higher-grade tumours, which may partially explain the reduced hormone receptor expression profiles. The loss of PR-IHC and ERPAS may reflect the involvement of EMT in lymphogenic locations or be related to dedifferentiation. It is known that dedifferentiation can occur in recurrent EC, leading to higher tumour grade and the loss of PR-IHV expression [34].

However, the loss of PR-IHC and ERPAS is not universally present in all metastases, as suggested by higher levels of PR-IHC and ERPAS in hematogenous and intra-abdominal locations. This may suggest that EMT is not the only mechanism driving metastases or that EMT occurs in these lesions without PR-IHC and ERPAS loss.

### 4.2. Radiotherapy and ER/PR-IHC and ERPAS

Both EBRT and VBT are frequently applied in EC as adjuvant therapy for local control [7,47] or with curative intent in isolated vaginal vault recurrences [48,49]. Due to the retrospective nature of this study and unclear radiation fields, this analysis mainly analysed vaginal vault biopsies. When comparing ER/PR-IHC and ERPAS between irradiated and non-irradiated biopsies, no significant differences were observed. There is limited literature on endometrial activity or changes in IHC after radiotherapy. Soslow et al. suggested that adjuvant RT does not alter hormone receptor expression in EC, which is consistent with the results of this study [34]. Furthermore, some residual functional endometrium was retained in cervical cancer treated with curative radiotherapy [50]. This implies that the application of HT can be considered either before or after RT, depending on ER/PR-IHC, which is also in line with our findings.

### 4.3. Strengths and Weaknesses

This study is the first analysis of ER/PR-IHC expression and ERPAS at different tumour locations in patients with advanced and recurrent disease. Inherent to the retrospective character of this study, there are some limitations that need to be addressed. As the metastatic and recurrent tumour biopsies were not matched to the primary tumour, comparisons within individual patients were not feasible.

Due to the inclusion criteria of the PROMOTE-study, in which only patients receiving hormonal therapy were eligible, this cohort represents a selection of women with advanced stage or recurrent EC. As a result, this cohort is characterised by relatively high ER/PR-IHC expression and a relatively higher number of EECs. Only 13% of the biopsies were high-grade tumours, compared to 40–50% reported in the literature [35]. The limited number of cases included in this study hindered the power of the subgroup analysis to draw significant conclusions. In addition, biopsies were taken from the most accessible tumour location and, therefore, a small number of samples from more difficult locations such as the lung, liver and lymph nodes could be included. However, this reflects daily clinical practice. Finally, variability in the diagnostic process, treatment and follow-up amongst patients further complicates interpretation. The relationship between ER/PR-IHC end ERPAS, tumour location and response to HT would have been a good addition to this study but was hampered by the RECIST criteria (Response evaluation criteria in solid tumors), which only described the target lesion. Data on the response of the specific biopsied location were, therefore, incomplete and not included in this study.

### 4.4. Clinical and Future Perspectives

The efficacy of hormonal therapy depends mainly on the percentage of PR-IHC expression in the tumour [19,24,25,26]. Therefore, it is important to acquire a biopsy from a metastatic site prior to initiating hormonal therapy to assess receptor status, as changes in receptor status may have occurred during the metastatic process. Based on the results of this study, it appears that lymphogenic metastases may have lower PR-IHC expression and ERPAS, potentially resulting in the lower efficacy of HT.

However, receptor status is not routinely assessed prior to the initiation of therapy. Typically, tumour biopsies are obtained from the most accessible location, often the uterus, in advanced EC. In lymphogenic locations, if HT is the preferred option, a lymphogenic biopsy is indicated, although this can be challenging due to anatomical difficulties (blood vessels, lymph nodes). In addition, the reliance on a single target lesion for the response assessment according to the RECIST guidelines prompts consideration of whether the biopsy should be obtained from this specific target lesion. Consideration should be given to obtaining biopsies from multiple locations, as a single biopsy may not reflect ER/PR-IHC and ERPAS from all metastases. This approach ensures the histological confirmation of advanced disease and clarifies potential tumour heterogeneity at metastatic sites, particularly in cases of lymphatic spread, where PR-IHC loss is more common. Less invasive methods, such as 18F-fluoroestradiol-PET (FES-PET), are also being investigated and may provide a solution in difficult biopsy locations, such as lymphogenic locations.

The prospective analysis of ER/PR-ICH and ERPAS by tumour location and their correlation with treatment response is essential to improve the selective use of hormonal therapy in advanced and recurrent EC. In addition, ERPAS has been shown to be useful in predicting response to hormonal therapy and in identifying hormone-driven tumour growth in metastatic lesions, justifying its inclusion alongside IHC assessment [24].

These recommendations will be incorporated into the PROMOTE-Prospective study, which aims to improve predictive biomarkers and patient selection in advanced and recurrent EC. By tailoring the use of hormonal therapy based on refined clinical and molecular criteria rather than solely on the experience of the treating physician, the use of hormonal therapy in EC can be optimised, resulting in greater cost-effectiveness.

## 5. Conclusions

In this study, an overall high expression of ER/PR and ERPAS was observed across different tumour locations. Comparing all tumour locations, there was a trend towards lower expression in lymphogenic locations compared to other locations. This potentially reflects the different mechanisms of cancer spread and may indicate reduced sensitivity to hormonal therapy. ER/PR-IHC expression was not significantly affected by tumour grade or previous radiotherapy. Larger studies are needed to confirm the differences in hormone receptor presence and activity between tumour locations and possible implications for endometrial cancer treatment.

## Figures and Tables

**Figure 1 cancers-16-02084-f001:**
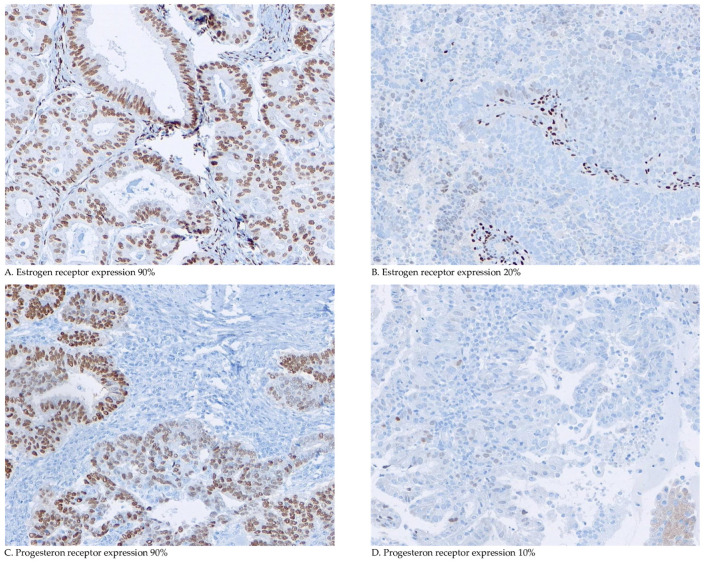
Microscopic immunohistochemical oestrogen and progesterone receptor staining and scoring. Representative examples of immunohistochemical analyses of oestrogen and progesterone receptor (ER/PR-IHC) and percentage of tumour cells expressing nuclear ERα and PR.

**Figure 2 cancers-16-02084-f002:**
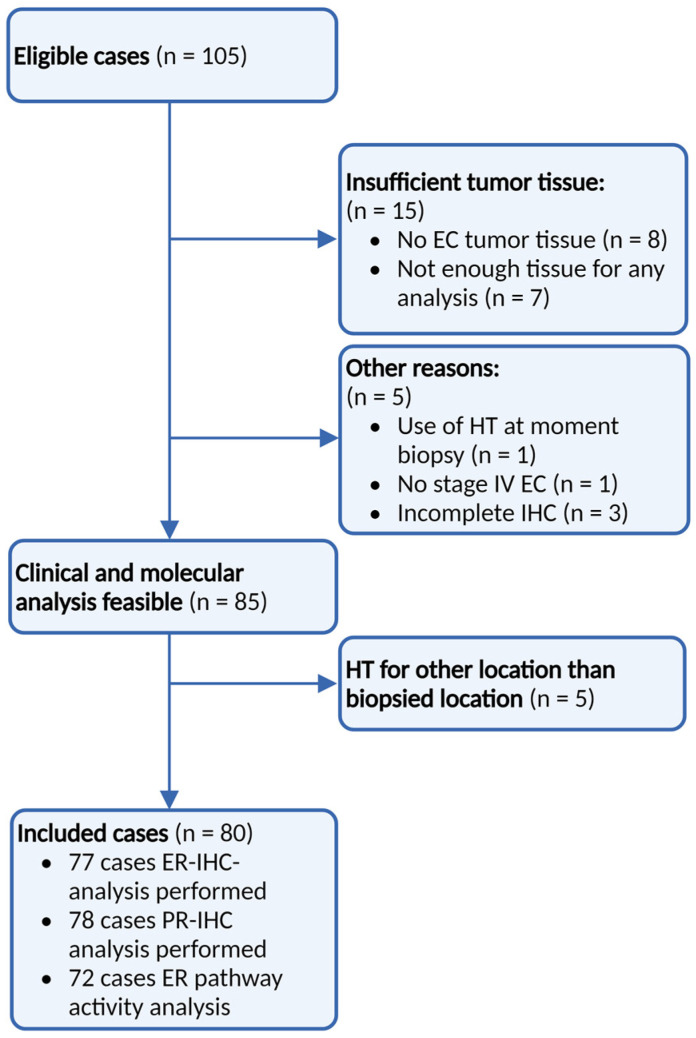
Patient inclusion. Identification of all women treated with any type of hormonal therapy for advanced and recurrent EC from 2012 up to 2016 using a retrospective search of hospital databases. ER: Oestrogen receptor, PR: progesterone receptor, IHC: immunohistochemistry, and HT: hormonal therapy.

**Figure 3 cancers-16-02084-f003:**
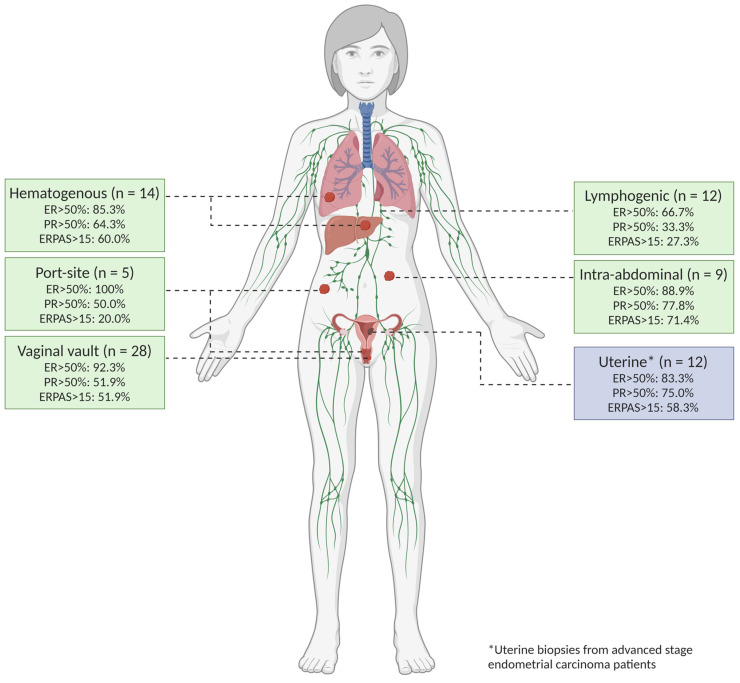
Tumour locations. All biopsied tumour locations were classified according to the hypothesised route of spread based on an article by Kurra et al. [40].

**Figure 4 cancers-16-02084-f004:**
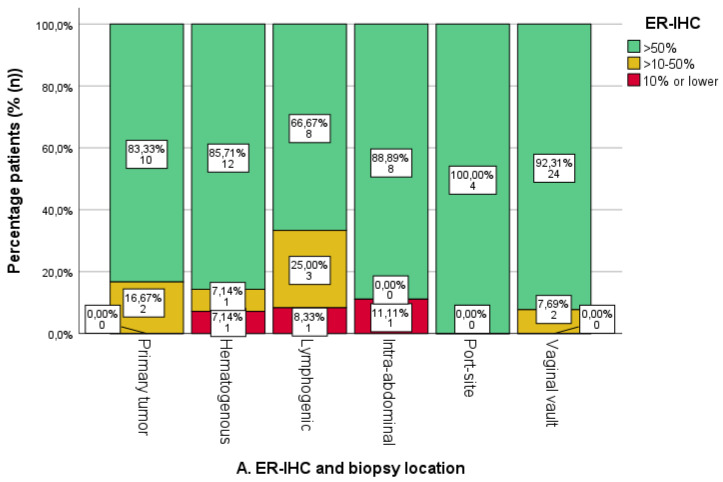
Oestrogen (ER) and progesterone receptor (PR) immunohistochemistry (IHC) categories and ER pathway activity score categories per sampled tumour location. The number and percentage of patients in each category are shown on the labels. (**A**). ER-IHC expression was categorised as ≤10%, 10–50%, and >50% in relation to tumour location. (**B**). PR-IHC expression categorised as ≤10%, 10–50%, and >50% in relation to tumour location. (**C**). ER pathway activity score with cut-off >15 (as the previously identified optimal cut-off value in the PROMOTE study [24]). * Uterine biopsies from patients with advanced-stage endometrial cancer.

**Figure 5 cancers-16-02084-f005:**
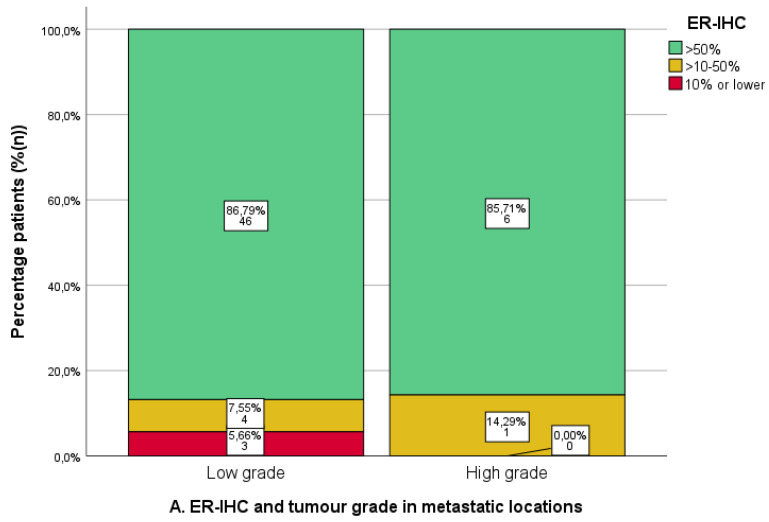
Immunohistochemical analysis of oestrogen (ER) and progesterone receptor (PR) immunohistochemistry (IHC) and ER -pathway activity score in metastatic locations in relation to tumour grade. Low grade: tumour grade 1 or 2. High grade: tumour grade 3. (**A**). ER-IHC expression categorised as ≤10%, 10–50%, and >50%. (**B**). PR-IHC expression categorised as ≤10%, 10–50%, and >50%. (**C**). ER pathway activity score with cut-off >15 (as the previously identified optimal cut-off value in the PROMOTE study [24]).

**Figure 6 cancers-16-02084-f006:**
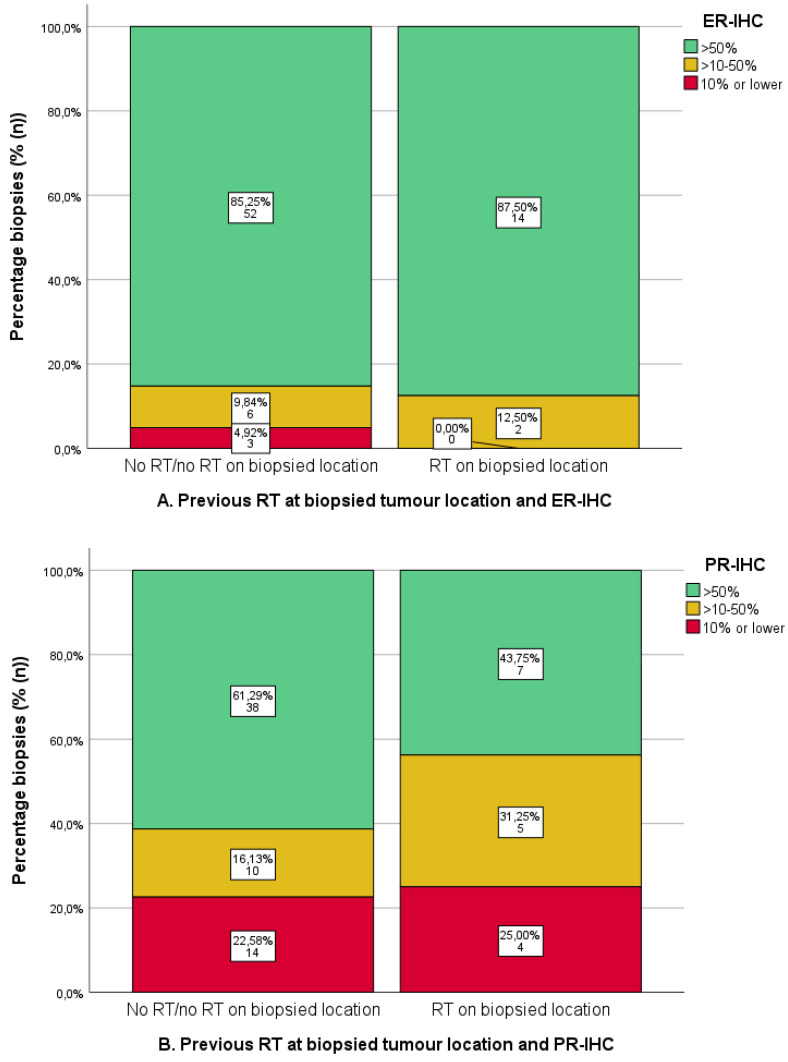
Immunohistochemistry for oestrogen receptor (ER-IHC) (**A**) and progesterone receptor (PR) (**B**) and oestrogen receptor pathway activity score (ERPAS) (**C**) in tumour locations with and without prior radiotherapy (RT). Predetermined cut-offs of ER-IHC and PR-IHC (>50%, >10–50%, ≤10%) and ERPAS > 15 are shown by the different categories (as the previously identified optimal cut-off value in the PROMOTE study [24]).

**Table 1 cancers-16-02084-t001:** Clinicopathological characteristics of included patients. SD: standard deviation, BMI: body mass index, EEC: endometrioid-type endometrial cancer, NEEC: non-endometrioid endometrial cancer, ER: oestrogen receptor (0–100%), PR: progesterone receptor (0–100%), ERPAS: oestrogen receptor pathway activity score (0–100%). Progestin therapy includes the following: medroxyprogesterone acetate and megestrol. Aromatase inhibitors include the following: letrozole, anastrazole and exemestane. *p*-value: <0.05 is considered significant. * Biopsies from exclusively advanced EC patients. ^a^ available for *n* = 57, ^b^ analysis for radiotherapy yes/no, ^c^ analysis for chemotherapy yes/no, ^d^ analysis for progestin vs. tamoxifen/aromatase inhibitor therapy.

	Total n^o^ of Cases *n* = 80	Uterine * *n* = 12 (15.0%)	Hematogenous *n* = 14 (17.5%)	Lymphogenic *n* = 12 (15.0%)	Intra-Abdominal *n* = 9 (11.3%)	Port-Site *n* = 5 (6.3%)	Vaginal Vault *n* = 28 (35.0%)	*p* Value
Age (CI)	71.9 (70.0–74.1)	73.7 (65.6–81.7)	73.1 (67.0–79.2)	70.8 (64.3–77.2)	67.2 (60.2–74.2)	71.0 (55.1–86.9)	72.6 (69.9–75.3)	0.650
BMI (CI)	30.4 (28.4–32.3)	29.3 (22.4–36.2)	32.8 (26.2–39.5)	29.2 (25.4–32.9)	33.5 (27.1–39.9)	30.2 (19.0–41.3)	28.7 (25.1–32.4)	0.639
Tumor type								
Recurrence	54 (67.5)	0 (0)	11 (78.6)	3 (25.0)	7 (77.8)	5 (100)	28 (100)	<0.001
Advanced	26 (32.5)	12 (100)	3 (21.4)	9 (75.0)	2 (22.2)	0 (0)	0 (0)	
Grade ^a^								
Grade 1–2 (%)	61 (76.3)	7 (58.3)	12 (85.7)	7 (58.3)	7 (77.8)	3 (60.0)	25 (89.3)	0.118
Grade 3 (%)	13 (16.3)	4 (33.3)	1 (7.1)	4 (33.3)	0 (0)	1 (20.0)	3 (10.7)	
Unknown (%)	6 (7.5)	1 (8.3)	1 (7.1)	1 (8.3)	2 (22.2)	1 (20.0)	0 (0)	
Histology								
EEC (%)	76 (93.8)	11 (91.7)	14 (100)	10 (83.3)	9 (100)	5 (100)	26 (92.9)	0.379
NEEC (%)	4 (5.0)	1 (8.3)	0 (0)	2 (16.7)	0 (0)	0 (0)	1 (3.6)	
Mixed type (%)	1 (1.3)	0 (0)	0 (0)	0 (0)	0 (0)	0 (0)	1 (3.6)	
Previous therapy								
Radiotherapy	33 (42.3)	0 (0)	10 (76.9)	2 (16.7)	2 (22.2)	3 (75.0)	16 (57.1)	<0.001 ^b^
Chemotherapy	6 (7.6)	0 (0)	0 (0)	1 (8.3)	2 (22.2)	0 (0)	3 (10.7)	0.346 ^c^
Drug type								
Progestin	61 (76.3)	9 (75.0)	13 (92.9)	9 (75.0)	6 (66.7)	2 (40.0)	22 (78.6)	0.622 ^d^
Tamoxifen	8 (10.0)	1 (8.3)	0 (0)	2 (16.7)	2 (22.2)	1 (20.0)	2 (7.1)	
Aromatase inhibitor	8 (10.0)	2 (16.7)	1 (7.7)	1 (8.3)	0 (0)	0 (0)	4 (14.3)	
Unknown	3 (3.8)	0 (0)	0 (0)	0 (0)	1 (11.1)	2 (40.0)	0 (0)	
ER expression (mean %, SD)	77.3 (24.8)	78.5 (23.2)	80.0 (29.2)	63.8 (31.7)	76.4 (30.0)	83.8 (16.0)	80.9 (17.7)	0.556
PR expression (mean %, SD)	49.5 (33.5)	54.2 (30.5)	52.9 (35.1)	34.3 (36.5)	63.9 (28.6)	37.5 (33.0)	49.5 (34.0)	0.578
ERPAS (mean %, SD)	16.2 (11.7)	20.3 (12.1)	16.3 (12.3)	11.0 (8.7)	20.7 (9.2)	17.2 (14.6)	15.2 (12.4)	0.406

## Data Availability

The raw data of this study can be made available upon reasonable request. Please contact the authors for requests.

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
