# Peer review of "Hormone Receptor Expression and Activity for Different Tumour Locations in Patients with Advanced and Recurrent Endometrial Carcinoma"

_cancers, 2024, doi:10.3390/cancers16112084_

Round 1

Reviewer 1 Report

Comments and Suggestions for Authors

This article describes the difference in hormone receptor expression in different locations in patients with advanced and recurrent endometrial cancer. The authors used immunohistochemistry along with an ER-pathway activity test using mRNA. Their study is interesting and challenging. However, there are several things to be considered, as shown below. 

1.        The authors investigated ER/PR status and ERPAS in detail. Their study is challenging. However, regrettably, they cannot take significant results from a small number. They need to increase the number of samples to obtain more impact.

2.        It is intriguing to know the chronological changes in ER/PR status and ERPAS.

3.        In figure 3, the title of figure 3B is missing.

Comments on the Quality of English Language

Minor editing of English language is recommended.

Author Response

Please look at the attached files for the reply.

Reviewer 2 Report

Comments and Suggestions for Authors

The manuscript “Hormone receptor expression and activity for different tumour locations in patients with advanced and recurrent endometrial carcinoma” presents exciting findings on ER/PR-IHC and ERPAS expression in different metastatic locations of endometrial cancer. It highlights a potential association between lower PR-IHC/ERPAS in lymphogenic spread and reduced hormonal therapy (HT) efficacy. This study emphasizes the importance of receptor assessment before initiating HT, especially obtaining biopsies from multiple sites, particularly for lymphatic spread. Here, the authors advocate for prospective studies correlating ER/PR-IHC/ERPAS with treatment response.

However, authors should represent the IHC microscopic images in the manuscript to support their claims. 

Overall, the authors raised valuable points about potential heterogeneity in ER/PR-IHC/ERPAS expression across metastatic sites. However, the limitations of the study design necessitate future investigation through prospective studies with larger cohorts.

The manuscript is impeccably composed and highly deserving of recommendation for publication after minor revisions regarding IHC data.

Author Response

Please look at the attachements for the reply to the review report.

Round 2

Reviewer 1 Report

Comments and Suggestions for Authors

This version of the article is fine. Although the sample size is relatively small, conclusions were drawn from the data analyzed.